# Poly(ester imide)s Possessing Low Coefficients of Thermal Expansion and Low Water Absorption (V). Effects of Ester-linked Diamines with Different Lengths and Substituents

**DOI:** 10.3390/polym12040859

**Published:** 2020-04-08

**Authors:** Masatoshi Hasegawa, Tomoaki Hishiki

**Affiliations:** Department of Chemistry, Faculty of Science, Toho University, 2-2-1 Miyama, Funabashi, Chiba 274-8510, Japan; tomoaki.hishiki@nitto.com

**Keywords:** poly(ester imide)s, heat resistance, coefficients of thermal expansion (CTE), water uptake, coefficients of hygroscopic (humidity) expansion (CHE), flame retardancy, dielectric constants, dissipation factors, flexible printed circuit boards (FPC)

## Abstract

A series of ester-linked diamines, with different lengths and substituents, was synthesized to obtain poly(ester imide)s (PEsIs) having improved properties. A substituent-free ester-linked diamine (AB-HQ) was poorly soluble in *N*-methyl-2-pyrrolidone at room temperature, which forced the need for polyaddition by adding tetracarboxylic dianhydride solid into a hot diamine solution. This procedure enabled the smooth progress of polymerization, however, accompanied by a significant decrease in the molecular weights of poly(amic acid)s (PAAs), particularly when using hydrolytically less stable pyromellitic dianhydride. On the other hand, the incorporation of various substituents (–CH_3_, –OCH_3_, and phenyl groups) to AB-HQ was highly effective in improving diamine solubility, which enabled the application of the simple polymerization process without the initial heating of the diamine solutions, and led to PAAs with sufficiently high molecular weights. The introduction of bulkier phenyl substituent tends to increase the coefficients of thermal expansion (CTE) of the PEsI films, in contrast to that of the small substituents (–CH_3_, –OCH_3_). The effects of ester-linked diamines, consisting of longitudinally further extended structures, were also investigated. However, this approach was unsuccessful due to the solubility problems of these diamines. Consequently, the CTE values of the PEsIs, obtained using longitudinally further extended diamines, were not as low as we had expected initially. The effects of substituent bulkiness on the target properties, and the dominant factors for water uptake (*W*_A_) and the coefficients of hygroscopic expansion (CHE), are also discussed in this study. The PEsI derived from methoxy-sustituted AB-HQ analog and 3,3′,4,4′-biphenyltetracarboxylic dianhydride achieved well-balanced properties, i.e., a very high *T*_g_ (424 °C), a very low CTE (5.6 ppm K^−1^), a low *W*_A_ (0.41%), a very low CHE value (3.1 ppm/RH%), and sufficient ductility, although the 26 μm-thick film narrowly missed certification of the V-0 standard in the UL-94V test. This PEsI film also displayed a moderate ε_r_ (3.18) and a low tan δ (3.14 × 10^−3^) at 10 GHz under 50% RH and at 23 °C. Thus, this PEsI system is a promising candidate as a novel dielectric substrate material for use in the next generation of high-performance flexible printed circuit boards operating at higher frequencies (≥10 GHz).

## 1. Introduction

The importance of heat-resistant polymers as electrical insulation materials has increased significantly in a variety of electrical and microelectronic applications. The most reliable high-temperature polymeric materials for these applications are aromatic polyimides (PIs). That is owing to their excellent combined properties, including short-term heat resistance (glass transition temperature, *T*_g_) and long-term heat resistance (thermal decomposition temperature, *T*_d_), flame retardancy, resistance to chemicals, mechanical properties, and dimensional stability against thermal cycles in the device fabrication processes [1,2,3,4,5,6,7,8,9,10]. One of the widest uses of PI products is, at present, their applications as dielectric substrates in flexible printed circuit boards (FPCs). However, more recently, there has been a demand for further improving the properties of dielectric substrates for their application in next-generation FPCs, to further increase the *T*_g_ and dimensional stability against water/moisture absorption and to further decrease water uptake, dielectric constants (*ε*_r_), and dissipation factors (tan *δ*) at higher operating frequencies (in GHz range). However, in fact, it is difficult to achieve these target properties simultaneously by simply combining conventional monomers for PIs (tetracarboxylic dianhydrides and diamines) and even by copolymerization using multiple monomers. Thus, new monomer development is strongly desired for present purposes.

To date, we have investigated aromatic poly(ester imide)s (PEsIs) as promising candidates for new dielectric substrate materials for FPC applications [11,12,13,14]. For example, longitudinal structural extension of ester-linked tetracarboxylic dianhydrides, as shown in Figure 1(left), was very effective in achieving excellent combined properties, i.e., a very high *T*_g_, an ultralow coefficient of thermal expansion (CTE), an extremely low coefficient of hygroscopic expansion (CHE), high flame retardancy, sufficient film ductility, and a low tan *δ* at 10 GHz [14]. These results motivated us to apply a similar structural modification approach to diamine monomers [Figure 1(right)]. However, there is a concern that longitudinally extended structures of ester-linked diamines may cause a significant decrease in their solubility, consequently making it difficult to apply the ordinary equimolar polyaddition process of [poly(amic acid)s (PAAs)], where tetracarboxylic dianhydride solid is added to a homogeneous diamine solution with a high solute content without heating. A possible strategy to avoid the diamine solubility problem is to incorporate an adequate substituent for maintaining excellent film properties into the ester-linked diamines. This idea was inspired by the prominent positive substituent effects observed in our previous studies [15,16,17].

In the present study, we used a series of ester-linked diamines with, and without, substituents and investigated the substituent effects on the target properties of the PEsIs [18]. The results show the potential for PEsIs, developed in this study, as new dielectric substrate materials for use in the next generation of high-performance FPCs.

## 2. Experimental Section

### 2.1. Materials

#### 2.1.1. Monomer Synthesis

In this study, a series of ester-linked diamines (AB-X) was synthesized using various hydroquinones and bisphenols [Appendix A] according to the reaction schemes shown in Figure 2. A typical synthetic procedure is as follows. 4-Nitrobenzoyl chloride (4-NBC, 20 mmol) was dissolved in anhydrous tetrahydrofuran (THF, 16.7 mL) in a flask sealed with a septum cap. In a separate sealed flask, methoxyhydroquinone (MeOHQ, 10 mmol) was dissolved in THF (6.3 mL) in the presence of pyridine (60 mmol, 4.9 mL) as an HCl acceptor. To the 4-NBC solution cooled at 0 °C, the MeOHQ solution was added slowly using a syringe with continuous magnetic stirring, after which the reaction mixture was stirred at room temperature for 12 h. The precipitate formed was collected by filtration, washed with a small quantity of THF and a large quantity of water, and vacuum-dried at 100 °C for 18 h (yield: 82%). The pale-yellowish product showed a sharp endothermic peak for melting at 238 °C on differential scanning calorimetry (DSC, Netzsch, DSC3100) conducted at a heating rate of 5 °C min^−1^. The analytical data are as follows. FT-IR (KBr plate method, cm^−1^): 1738 (ester, C=O), 1526/1348 (NO_2_), 1510 (1,4-phenylene), 1269 (ester + methoxy, O–C). ^1^H-NMR [400 MHz, dimethyl sulfoxide (DMSO)-*d*_6_, *δ*, ppm]: 8.48–8.39 [m, 8H (relative integrated intensity: 7.97H), 2,6- + 2′,6′- + 3,5- + 3′,5′-protons of the nitrobenzene units], 7.45 [d, 1H (0.97H), *J* = 7.8 Hz, 6-proton of the central 2-methoxyhydroquinone (MeOHQ) unit], 7.34 [s, 1H (0.95H), 3-proton of MeOHQ], 7.06–7.03 [m, 1H (0.97H), 5-proton of MeOHQ], 3.79 [s, 3H (3.00H), OCH_3_].

A portion of the dinitro compound obtained (4.56 mmol) was dissolved in *N*,*N*-dimethylformamide (DMF, 30 mL) in the presence of Pd/C (0.067 g) as a catalyst. The reaction mixture was refluxed at 100 °C for 5 h in a hydrogen atmosphere. The progress of the catalytic reduction was monitored by thin-layer chromatography. After the catalyst residue was removed by hot filtration, the filtrate was concentrated with an evaporator. The precipitate obtained was washed with THF and dried at 160 °C for 12 h under vacuum. The crude product obtained was recrystallized from 1,4-dioxane, collected by filtration, and dried at 140 °C for 12 h under vacuum. Pale-brown needle crystal was obtained with a yield of 85% and characterized. FT-IR (KBr plate method, cm^−1^): 3482/3378 (amine, N–H), 1707 (ester, C=O), 1518 (1,4-phenylene), 1285 (ester + methoxy, O–C). ^1^H-NMR (400 MHz, DMSO-*d*_6_, *δ*, ppm): 7.82–7.77 [m, 4H (4.00H), 3,5- + 3′,5′-protons of the terminal aniline (AN) unit], 7.18 [d, 1H (0.97H), *J* = 8.3 Hz, 6-proton of the MeOHQ unit], 7.03 [sd, 1H (1.00H), *J* = 2.7 Hz, 3-proton of MeOHQ], 6.80 [dd, 1H (1.00H), *J* = 8.5, 2.7 Hz, 5-proton of MeOHQ], 6.65–6.62 (m, 4H (4.00H), 2,6- + 2′,6′-protons of the AN unit], 6.20, 6.18 [s, 4H (3.95H), NH_2a_ + NH_2b_], 3.75 [s, 3H (3.00H), OCH_3_]. Elemental analysis: Calcd. (%) for C_21_H_18_O_5_N_2_ (378.38 g mol^−1^): C, 66.66; H, 4.79; N, 7.40, Found: C, 66.47; H, 4.77; N, 7.00. Melting point (DSC): 254 °C. These data confirm that the product is the desired diamine (AB-MeOHQ). The other AB-Xs (Figure 2, Appendix A) were synthesized using different raw materials (Appendix A) and characterized in a similar manner.

#### 2.1.2. Common Monomers

The structures and abbreviations of common monomers used in this study are shown in Figure 3. Their commercial sources, pre-treatment conditions, and melting points are listed in Appendix A.

#### 2.1.3. Polymerization and Thermal Imidization for PEsI Film Preparation

PAAs were prepared by equimolar polyaddition of tetracarboxylic dianhydrides and diamines according to the scheme shown in Figure 4. A typical procedure is as follows. A diamine (3 mmol) was dissolved in anhydrous *N*-methyl-2-pyrrolidone (NMP) in a sealed bottle. Then, tetracarboxylic dianhydride powder (3 mmol) was added to the diamine solution with continuous magnetic stirring. The initial total solid content was 15 or 20 wt%. The reaction mixture was stirred at room temperature for typically 96 h in the sealed bottle to obtain a homogeneous solution. If necessary, the reaction mixture was gradually diluted repeatedly with minimal quantity of the same solvent to ensure effective magnetic stirring. When a minor portion of insoluble matters remained in the PAA solutions after the polymerization, they were removed using a glass filter.

The formation of PAAs was confirmed by transmission-mode FT-IR spectroscopy (Jasco, Tokyo, Japan, FT/IR 4100 infrared spectrometer) using separately prepared thin cast films (4–5 μm thick) with a non-uniform thickness to erase the interference fringes. A typical FT-IR spectrum of the PAAs is shown in Figure 5a. The spectrum shows specific bands (cm^−1^): 3312 (amide, N–H), 2623 (hydrogen-bonded carboxylic acid, O–H), 1721 (ester, C=O), 1599 (biphenyl group), 1532 (amide, C=O), and 1256 (ester + methoxy, O–C). The spectrum does not contain the IR bands corresponding to the monomers (e.g., the C=O stretching band at 1860 cm^−1^ for the acid anhydride groups of the tetracarboxylic dianhydride), indicating that the equimolar polyaddition occurred quantitatively.

The PEsI films were prepared via the conventional thermal imidization process as follows. The PAA solution was bar-coated on a glass substrate and dried at 80 °C for 3 h in an air-convection oven. In some cases where a turbid PAA solution was obtained after the polymerization (e.g., in the PMDA/AB-HQ system, see the Section 3.1.1), the PAA solution was homogenized by heating at 100 or 120 °C for 1 min and promptly cast at 120 °C for 2 h to obtain clear films. The PAA films were imidized at 250 + 300 °C for each 1 h on the substrate under vacuum, and, finally, annealed at 350 °C for 1 h under vacuum without the substrate to remove residual stress. In some cases, these thermal conditions were adjusted to obtain better-quality PEsI films.

Complete imidization was also confirmed from the FT-IR spectra of separately prepared thin films, as shown in Figure 5b, on the basis of the appearance of specific bands (cm^−1^): 3077 (C_Ar_–H), 1777 (imide, C=O), 1721 (imide + ester, C=O), 1609 (biphenyl group), 1512 (*p*-phenylene), 1370 (imide, N–C_Ar_), 1260 (ester + methoxy, O–C), and 737 (imide, ring deformation). In addition, no infrared bands characteristic of PAAs [e.g., the amide C=O stretching band at ~1660/1530 cm^−1^ and the carboxylic acid O–H stretching broad at ~2620 cm^−1^] were observed in this spectrum.

In this paper, the chemical compositions of PEsI systems are denoted using the abbreviations of the monomer components used, i.e., tetracarboxylic dianhydride, (A) and diamine, (B) as A/B for homopolymers and A/B1; B2 for copolymers.

### 2.2. Measurements

#### 2.2.1. Inherent Viscosities

The reduced viscosities (*η*_red_) of PAAs were measured in NMP at a solid content of 0.5 wt% at 30 °C using an Ostwald viscometer. The polyelectrolyte effect of the PAAs prevents the determination of the inherent viscosities (*η*_inh_) by extrapolation to zero concentration. Therefore, the *η*_red_ values at 0.5 wt% for PAAs are often regarded practically as the *η*_inh_ values. 

#### 2.2.2. Linear Coefficients of Thermal Expansion (CTE)

The CTE values of the PEsI films in the glassy region were measured as the averages between 100 and 200 °C [specimen size: 20 mm long (chuck-to-chuck distance: 15 mm), 5 mm wide, and typically 20 μm thick] by thermomechanical analysis (TMA). The measurements were carried out at a heating rate of 5 °C min^−1^ on a thermomechanical analyzer (Netzsch, Tokyo, Japan, TMA 4000) with a fixed load (0.5 g per unit film thickness in μm, i.e., 10 g load for 20 μm-thick films) in a dry nitrogen atmosphere. After the preliminary heating run to 120 °C and successive cooling to room temperature under a continuous dry nitrogen flow in the closed TMA chamber, the data were collected from the second heating run to remove the influence of adsorbed water.

#### 2.2.3. Linear Coefficients of Hygroscopic (Humidity) Expansion (CHE)

The CHE values of the PEsI films were determined at room temperature by monitoring the dimensional changes of the specimens maintained at a relative humidity (RH) of 80%. The measurements were carried out by connecting a precision humidity generator (Shinyei Technology, Tokyo, Japan, SRG-1R-1) to the above-mentioned TMA instrument. To prevent moisture absorption of the specimens occurring before the measurements, the TMA chuck parts were attached to both ends of the fixed-size specimens in advance and dried at 100 °C in a vacuum oven to remove absorbed water. Then, the chuck-attached specimens were promptly set in the TMA chamber, and maintained in dry nitrogen before the measurements. To avoid dew formation in the chamber, the relative humidity of the injected wet air was gradually increased from 50% to 70% RH in 10% RH intervals and held for 5 min at each step, finally being maintained at 80% RH. The dimensional changes of the specimens were monitored until the elongation–time curve was practically flat (typically, 15–18 h).

#### 2.2.4. Glass Transition Temperatures

The storage modulus (*E*′) and loss energy (*E*″) of the PEsI films were measured by dynamic mechanical analysis (DMA) at a heating rate of 5 °C min^−1^ and a sinusoidal load frequency of 0.1 Hz with an amplitude of 15 gf in a nitrogen atmosphere using the above-mentioned TMA instrument. The *T*_g_ values of the PEsI films were determined from the peak temperature of the *E*″ curve.

The thermal stabilities of the PEsI films were evaluated from the 5% weight loss temperatures (*T*_d_^5^) by thermogravimetric analysis (TGA) on a thermo-balance (Netzsch, Tokyo, Japan, TG-DTA2000). TGA measurements were carried out at a heating rate of 10 °C min^−1^ in a dry nitrogen and air atmosphere. The small weight loss due to the desorption of water from the samples at around 100 °C in the TGA curves was compensated by an off-set at 150 °C to 0% weight loss for the data analysis.

#### 2.2.5. Water Absorption

The degrees of water absorption (*W*_A_, %) of the PEsI films were determined according to the JIS K 7209 standard using the Equation (1),
*W*_A_ = [(*W* − *W*_0_)/*W*_0_] × 100(1)
where *W*_0_ is the weight of a film sample just after vacuum-drying at 50 °C for 24 h, and *W* is the weight of the film immersed in water at 23 °C for 24 h, and carefully blotted dry with tissue paper. In this case, large film specimens (> 0.1 g) were used to minimize the experimental error.

To investigate the influence of the imide group (O=C–N–C=O) on the *W*_A_, the imide contents (*C*_i_, in wt%) were calculated using Equation (2),
*C*_i_ = *F*_W_ (imide)/*F*_W_ (unit) × 100(2)
where *F*_W_ (imide) and *F*_W_ (unit) denote the formula weights of the imide group, and the repeating units, respectively.

#### 2.2.6. Birefringence

The thickness-direction birefringence (Δ*n*_th_ = *n_xy_* – *n_z_*) of the PEsI films, which represents the relative extents of chain alignment in the *X*–*Y* direction, was obtained from the in-plane (*n*_xy_) and out-of-plane (*n*_z_) refractive indices. These were measured at 589.3 nm (sodium lamp, *D*-line) on an Abbe refractometer (Atago, Tokyo, Japan, 4T, *n*_D_ range: 1.47–1.87), equipped with a polarizer by using a contact liquid (sulfur-saturated methylene iodide *n*_D_ = 1.78–1.80), and a test piece (*n*_D_ = 1.92).

#### 2.2.7. Mechanical Properties

The tensile modulus (*E*), tensile strength (*σ*_b_), and elongation at break (*ε*_b_) of the PEsI films were measured using specimens with a fixed size (30 mm long, 3 mm wide, typically 20 μm thick) that was free of any defects, such as fine bubbles, using a mechanical testing machine (A & D, Tokyo, Japan, Tensilon UTM-II) at a cross head speed of 8 mm min^−1^ at room temperature. The data analysis was carried out using a data processing program (Softbrain, Tokyo, Japan, UtpsAcS Ver. 4.09) and averaged for the valid run numbers (typically, *n* > 15).

#### 2.2.8. Flame Retardancy

The flame retardancy of PEsI films was evaluated according to the UL-94 vertical burning test, and a set of five specimens was tested using fixed-size film samples [125 mm long, 13 mm wide, average thickness (*d*): ca. 20 μm]. When all the five specimens did not burn up to the clamped top edge, the samples are certified as having passed the V-0 standard.

#### 2.2.9. Dielectric Constants and Dissipation Factors

The dielectric constants (*ε*_r_) and dissipation factors (tan *δ*) in the *X*–*Y* direction for the PEsI films were measured at a frequency of 10 GHz under 50% RH and at 23 °C using a cavity resonator perturbation technique (IEC 62810) using PNA-L network analyzer (Agilent Technologies, Tokyo, Japan, N5230A) and a cavity resonator (Kanto Electronic Application and Development Inc., Tokyo, Japan, CP531). The measurements were performed by DJK Corp. as a contracted service. The data were calibrated with a reference sample (polytetrafluoroethylene, *ε*_r_ = 2.01, tan *δ* = 2.3 × 10^−4^)

## 3. Results and Discussion

### 3.1. PEsIs Derived from AB-HQ without Substituents

#### 3.1.1. Polymerizability of AB-HQ

AB-HQ was not sufficiently soluble in common amide solvents (NMP, DMAc), at room temperature, in order to begin the PAA polymerization without heating. Therefore, tetracarboxylic dianhydride powder was added to a hot NMP solution of AB-HQ to ensure the initial diamine solution homogeneity as precipitation occurs as soon as the hot AB-HQ solution is cooled to room temperature. This procedure enabled smooth progress of PAA polymerization. The *η*_inh_ values of the PAAs obtained are listed in Table 1. The equimolar polyaddition of pyromellitic dianhydride (PMDA) and AB-HQ (**system #1**) led to a viscous PAA solution with turbidity, which has a somewhat low *η*_inh_ value of 0.58 dL g^−1^. The turbidity probably arises from partial precipitation of the PAA formed on the basis of its very rigid chain structure. This PAA solution can be easily homogenized temporally upon heating at 120 °C for 1 min. The relatively low *η*_inh_ value obtained, which was lower than 1 dL g^−1^ as an empirical criterion for representing sufficiently high molecular weights of PAAs, was probably related to the fact that PMDA readily underwent ring-opening hydrolysis, particularly at elevated temperatures, which broke the strict equimolar condition of the monomers fed.

On the other hand, the use of 3,3′,4,4′-biphenyltetracarboxylic dianhydride (s-BPDA, **#2**), which is empirically much more stable to hydrolysis than PMDA, led to a significantly enhanced *η*_inh_ value (1.72 dL g^−1^), suggesting the formation of PAA with a sufficiently high molecular weight. This result suggests that AB-HQ has an essentially high polymerization reactivity with tetracarboxylic dianhydrides, one that is comparable to that of common aromatic diamines such as 4,4′-oxydianiline (4,4′-ODA). This assumption is also supported by the fact that the reaction of s-BPDA and 4,4′-ODA gave a PAA with a high *η*_inh_ value of 2.00 dL g^−1^ [19], which approximated that of the s-BPDA/AB-HQ system mentioned above. 

#### 3.1.2. Film Properties

The properties of the AB-HQ-based PEsI films are summarized in Table 1. The PMDA/AB-HQ system (**#1**) exhibited a very low CTE (3.6 ppm K^−1^) without showing a distinct glass transition up to 450 °C in the DMA curve, as is often observed in rod-like PI systems (e.g., PMDA/*p*-phenylenediamine (*p*-PDA) [20] and PMDA/2,2′-bis(trifluoromethyl)benzidine (TFMB) [21]). The absence of a distinct *T*_g_ on DMA also represents that this PEsI had the excellent dimensional stability over the very wide temperature range. The PEsI film (**#1**) also exhibited a high tensile modulus (*E* = 6.52 GPa). The observed low-CTE and high-modulus behavior results from a combined effect of a chemical factor, i.e., its rigid and linear main chain structure, and a physical factor, i.e., significant chain alignment along the film plane (*X*–*Y*) direction (called “in-plane orientation”), corresponding to a high Δ*n*_th_ value (0.13). The in-plane orientation behavior occurs during the thermal imidization of the PAA film fixed on the substrates [22,23,24,25]. However, as predicted, the PEsI film (**#1**) was not highly tough (*ε*_b_ < 10%), probably owing to insufficient chain entanglement, based on its very rigid chain structure. This system (**#1**) also showed good thermal stability, as suggested by the relatively high *T*_d_^5^ value (526 °C) in a nitrogen atmosphere.

Similar low-CTE and high-modulus behavior was also observed in the s-BPDA/AB-HQ system (**#2**). Another prominent feature of the PEsI is the extremely low CHE (0.5 ppm/RH%), which indicates very high dimensional stability against moisture absorption. On the other hand, the use of TA-HQ (**#3**) as the tetracarboxylic dianhydride was effective in reducing water uptake (*W*_A_ = 0.42%) and improving film ductility (*ε*_b_^max^ = 20.1%), although the CTE value was somewhat increased, compared to those of the other AB-HQ-based PEsIs (**#1** and **#2**). The significantly reduced *W*_A_ of the PEsI (**#3**) was closely related to the decreased content of highly polarized imide C=O groups, i.e., *C*_i_ = 18.2 wt% for TA-HQ/AB-HQ (**#3**), compared to *C*_i_ = 26.4 wt% for PMDA/AB-HQ (**#1**).

The s-BPDA/AB-HQ (**#2**) system was modified by copolymerization (**#4, #5**) using a flexible diamine (4,4′-ODA), although there was concern for an undesirable significant increase in the CTE. However, this approach improved significantly the film toughness (e.g., *ε*_b_^max^ = 47.1% for **#5**) while maintaining a low CTE property (e.g., 11.7 ppm K^−1^ for **#5**), which is lower than the CTE of copper foils (17–20 ppm K^−1^), as shown in Table 1. These copolymers also exhibited relatively high *T*_d_^5^ in N_2_, low *W*_A_, and significantly suppressed CHE values (e.g., 4.5 ppm/RH% for **#5**), which were much lower than those of conventional PIs (e.g., CHE ~ 20 ppm/RH% for a commercially available KAPTON-H film [14,26]).

#### 3.1.3. Isomer Effects

AB-HQ was compared with an isomer of AB-HQ, i.e., AB-RC (Figure 2, Appendix A), which consists of a distorted structure. In contrast to the ultralow-CTE behavior observed in the s-BPDA/AB-HQ (**#2**), the isomeric s-BPDA/AB-RC system showed no low CTE characteristics (52.5 ppm K^−1^), corresponding to the poor in-plane orientation of the latter system, as suggested by a low Δ*n*_th_ value (0.013). The results emphasize how important the overall chain linearity is for inducing significant in-plane orientation during thermal imidization, consequently generating low CTE characteristics.

Figure 6 displays the comparisons of the properties for the AB-HQ-based PEsIs and the counterparts derived from an isomeric ester-linked diamine, bis(4-aminophenyl)terephthalate (BPTP) [13], which has a different connecting order of the ester groups but a structural rigidity similar to that of AB-HQ. As shown in Figure 6a, AB-HQ tends to exhibit lower η_inh_-based polymerizability, compared to BPTP. This is probably attributed to the electron-withdrawing effect of the C=O groups located at the *para*-positions of the NH_2_ functional groups of AB-HQ, which reduces the basicity of the functional groups. However, this effect is much reduced for BPTP as the C=O groups are not directly connected to the terminal aniline units. 

On the other hand, no clear trend was observed in the comparison of CTE for the AB-HQ- and BPTP-based systems, as shown in Figure 6b, although considerably low CTE values were still maintained in both systems. An advantage of the AB-HQ-based PEsIs is observed in Figure 6c; they have lower *W*_A_ values than the BPTP-based counterparts, despite the same *C*_i_ value for these isomeric systems. These results suggest that the water uptake is dominated by not only *C*_i_ but also other secondary factors [14]. 

It is quite possible that the solubility of AB-HQ and the properties of AB-HQ-based PEsIs can be further improved by incorporation of adequate substituents into AB-HQ. This strategy also has great flexibility, since hydroquinone analogs with various substituents as the raw materials are extensively available [16], in contrast to the limited availability of substituted terephthalic acids for modifying BPTP.

### 3.2. PEsIs Derived from AB-HQ Analogs with Small Substituents

Table 2 summarizes the properties of PEsIs derived from methyl-substituted AB-HQ, i.e., AB-MHQ (Figure 2). The introduction of methyl group was effective in improving the solubility in NMP at room temperature and enabled the application of the common polymerization procedure without the initial heating of the diamine solution. Consequently, a PAA with a high η_inh_ value (1.21 dL g^−1^) was obtained easily, even when hydrolytically less stable PMDA was used.

The PMDA/AB-MHQ (**#6**) system, as well as s-BPDA/AB-MHQ (**#7**), also exhibited near zero CTE (–0.6 ppm K^−1^) and a considerably high tensile modulus (8.00 GPa), together with inevitably insufficient toughness, as listed in Table 2, which are features similar to those of the non-substituted counterparts (Table 1). On the other hand, the TA-HQ/AB-MHQ film (**#8**) was sufficiently tough (*ε*_b_^max^ = 34.4%) despite the absence of flexible ether linkages in the main chains, while maintaining a dramatically reduced *W*_A_ (0.23%) and a low CTE (15.5 ppm K^−1^), close to that of copper foil. Thus, there were no negative influences of this modification on the target properties, except for an appreciable decrease in the *T*_d_^5^, which is inevitable due to the presence of thermally less stable methyl side groups in the structure.

The methoxy substitution to AB-HQ also enhanced the diamine solubility in NMP at room temperature; as a result, the polymerization proceeded smoothly without the initial heating of the diamine solution and easily led to homogeneous solutions of PAAs with high η_inh_ values. The properties of AB-MeOHQ-based PEsIs are summarized in Table 3. The use of rigid tetracarboxylic dianhydrides [PMDA (**#9**) and s-BPDA (**#10**)] led to the PEsI films having considerably low CTE and very high modulus, which are very similar features to those of the corresponding non-substituted and methyl-substituted counterparts (Table 1 and Table 2). In the s-BPDA/AB-MeOHQ system (**#10**), an attractive feature was observed: a significantly improved ductility (*ε*_b_^max^ = 23%), compared to those of the s-BPDA-based counterparts (**#2, #7**). A similar substituent-assisted toughening effect was also observed in other PEsI systems [13,14] that we have reported previously, which can be explained by a chain slippage mechanism [27,28]. In addition, this PEsI had very low CTE and CHE values. Thus, the s-BPDA/AB-MeOHQ system (**#10**) exhibited excellent combined properties. This system was modified by copolymerization using 4,4′-ODA (30 mol%), and the resultant copolymer (**#12**) also achieved well-balanced properties, i.e., a very high *T*_g_, a very low CHE, a low CTE, and good toughness.

### 3.3. PEsIs Derived from AB-HQ Analogs with Bulky Substituents

The properties of PEsIs derived from AB-PhHQ, which includes a bulky phenyl side group, are summarized in Table 4. In the PMDA/AB-PhHQ system (**#13**), no distinct glass transition was observed on DMA, as in the other PMDA-based counterparts (**#1, #6, #9**). However, when using s-BPDA (**#14**) and TA-HQ (**#15**) as the tetracarboxylic dianhydrides, each glass transition became detectable, in contrast to the cases in the counterparts with the small substituents (CH_3_ and OCH_3_ groups). These results probably reflect the fact that close chain stacking was disturbed by the bulky phenyl side groups, and, consequently, sufficient molecular motions were allowed above each *T*_g_.

The PMDA/AB-PhHQ (**#13**) and s-BPDA/AB-PhHQ (**#14**) systems also maintained very low CTE values. However, the TA-HQ/AB-PhHQ film (**#15**) resulted in an increased CTE (25.8 ppm K^−1^). In addition, the s-BPDA/AB-PhHQ-based copolymer (**#16**) using 4,4′-ODA (30 mol%) failed to maintain low CTE characteristics. Thus, the introduction of a bulky phenyl side group contributes to the deterioration of low CTE property in some cases.

AB-14DHN (Figure 2), which includes a different type of side group, was also examined in this study. In this diamine, the spin motion of the benzene side group is inhibited because it is condensed to the HQ unit, in contrast to that of the phenyl side group in AB-PhHQ. In this regard, there is similarity for the side groups of AB-14DHN and cardo-type monomers, where the fluorenyl side group bound to the main chains via an *sp*^3^ carbon atom is not spin-rotatable [29,30,31,32]. Therefore, the AB-14DHN-based systems are expected to show some properties characteristic of the cardo-type PIs, for example, an enhanced *T*_g_. The properties of the AB-14DHN-based PEsIs are summarized in Table 5. The PMDA/AB-14DHN (**#17**) and s-BPDA/AB-14DHN (**#18**) systems showed CTE and moduli comparable to those of the counterparts using AB-PhHQ. However, when using TA-HQ, the difference in the properties between the AB-14DHN-based and AB-PhHQ-based systems became clear: i.e., the former (**#19**) still maintains a low CTE (16.1 ppm K^−1^), in contrast to the increased CTE (25.8 ppm K^−1^) for the latter (**#15**). There is a similar situation in the comparison of the CTE values for the AB-14DHN-based (**#20**) and AB-PhHQ-based copolymers (**#16**). Thus, there is almost no negative influence of the condensed benzene side group in AB-14DHN on the low CTE, unlike the phenyl side group in AB-PhHQ. In addition, the AB-14DHN-based systems, as expected, showed higher *T*_g_s than those of the AB-PhHQ-based counterparts.

### 3.4. PEsIs Derived from Ester-linked Diamines with Longitudinally further Extended Structures

The use of AB-44BP (Figure 2), which consists of a rigid and longitudinally further extended structure, is expected to contribute to further decreases in the *W*_A_ and CHE of the PEsI films, while maintaining low CTE characteristics, owing to the combination of the expected rigid/linear chain structure and the decreased content of the highly polarized imide groups. The properties of AB-44BP-based systems are summarized in Table 6. The equimolar polyaddition of AB-44BP and PMDA resulted in a PAA with a relatively low η_inh_ value (0.51 dL g^−1^). This is attributed to the insufficient solubility of AB-44BP, which forced heating the AB-44BP solution to homogenize it prior to the addition of PMDA powder. The resultant PEsI film (**#21**) showed no distinct *T*_g_ on DMA, as in the PMDA/AB-HQ system (**#1**), which reflects the rigid chain structure of the former. As expected, the PEsI film (**#21**) showed a significantly reduced *W*_A_ (0.48%) and a comparably high *T*_d_^5^ (in N_2_), compared to the AB-HQ-based counterpart (**#1**). However, contrary to our expectations, an undesirable increase in CTE (26.3 ppm K^−1^) was observed in this film (**#21**). This result is very likely related to the insufficient molecular weight of the PAA, which often acts disadvantageously in inducing prominent in-plane chain orientation during thermal imidization, as discussed regarding the mechanism in our previous studies [13,33]. Similar features were also observed in the related systems using the other tetracarboxylic dianhydrides [s-BPDA (**#22**) and TA-HQ (**#23**)].

AB-DP44BP (Figure 2) including two phenyl substituents was used in the next step. As expected, this modification made it sufficiently soluble in NMP, at room temperature, for conducting the ordinary polymerization process without the initial heating of the diamine solution. Therefore, a PAA with a very high η_inh_ value (2.05 dL g^−1^) was obtained easily, even when hydrolytically less stable PMDA was combined with this diamine. Table 7 summarizes the properties of AB-DP44BP-based PEsI films. Despite the introduction of the bulky phenyl substituents, which tends to increase the CTE as mentioned above, the PMDA/AB-DP44BP film (**#24**) exhibited a much lower CTE (11.2 ppm K^−1^) than the non-substituted counterpart (**#21**), probably reflecting the positive effect of the enhanced molecular weight. The use of s-BPDA (**#25**) as the tetracarboxylic dianhydride was suitable for reducing the *W*_A_ further, while maintaining the low CTE property. However, in the combination of AB-DP44BP and TA-HQ (**#26**), a negative influence of the bulky phenyl substituents emerged; the CTE was significantly increased. The s-BPDA/AB-DP44BP system was also modified using 4,4′-ODA. However, this approach (**#27**) failed to significantly improve film toughness while maintaining a low CTE property, as shown in Table 7.

We also used AB-MHQHB (Figure 2, Appendix A), which consists of a rigid and longitudinally further extended structure with a methyl side group, with expectations of further improvement in *W*_A_ and CHE, while maintaining sufficient diamine solubility and a low CTE property of the resultant PEsI films. However, this diamine was not highly soluble in NMP at room temperature. Therefore, PAA polymerization was carried out by adding tetracarboxylic dianhydride powder to a hot NMP solution of the diamine. The combination of AB-MHQHB and PMDA provided a PAA with a very low η_inh_ value (0.23 dL g^−1^), whereby the PAA cast film, as well as the corresponding PEsI film, was very brittle. On the other hand, the combination of s-BPDA and this diamine led to a PAA with an increased η_inh_ value (0.79 dL g^−1^). However, the properties of the resultant PEsI film were not so remarkable (CTE = 29.8 ppm K^−1^, *W*_A_ = 0.63%, CHE = 6.7 ppm/RH%, and *ε*_b_^max^ = 6.6%). Thus, the use of AB-MHQHB with the most longitudinally extended structure among the diamines, examined in this work, was not always effective in meeting targets, i.e., simultaneous achievement of low CTE (< ~15 ppm K^−1^), very low *W*_A_ (< ~0.3%), very low CHE (< ~3 ppm/RH%), and good toughness (*ε*_b_^max^ > ~20%). Overviewing, the PEsIs investigated in this study, some of the AB-MeOHQ-based and AB-MHQ-based PEsIs, derived from a controlled length of the ester-linked diamines, have well-balanced properties, compared to the PEsIs obtained using the diamines with longitudinally more extended structures (AB-44BP, AB-DP44BP, and AB-MHQHB).

### 3.5. Data Analysis

#### 3.5.1. Influence of Side Group Bulkiness on *T*_g_ and CTE

Figure 7a shows the comparison of *T*_g_ values for the PEsIs derived from three rigid tetracarboxylic dianhydrides and AB-HQ analogs with different substituents. Here, the column charts are arranged in order of side group bulkiness. As expected, The *T*_g_ tended to decrease with decreasing structural rigidity of the tetracarboxylic dianhydride components regardless of the substituents in the following order: PMDA > s-BPDA > TA-HQ. In the PMDA-based systems, side group bulkiness did not influence *T*_g_. On the other hand, an appreciable substituent effect on decreasing *T*_g_ was observed in the TA-HQ-based systems. This undesirable effect was much smaller when using s-BPDA as the tetracarboxylic dianhydrides.

A similar comparison for CTE values is shown in Figure 7b. As expected, CTE increased with decreasing structural rigidity of tetracarboxylic dianhydrides in the following order: PMDA < s-BPDA < TA-HQ. A clear trend is observed in the TA-HQ-based systems; CTE increases with increasing side group bulkiness, although this undesirable effect is significantly suppressed when using more rigid PMDA and s-BPDA. Thus, in some cases, a highly bulky side group, such as a phenyl group in PEsIs disturbs prominent in-plane chain orientation during the thermal imidization.

#### 3.5.2. Influence of the Position of Ester-linked Aromatic unit on Properties

The PMDA/AB-HQ system (**#1**), which includes the ester-linked hydroquinone dibenzoate (HQDB) unit (Ar-COO-Ar-OCO-Ar) in the diamine structure (B-type), was compared with another type of PEsI derived from TA-HQ (Figure 3) and *p*-PDA, which includes the HQDB unit in the tetracarboxylic dianhydride structure (A-type). Our previous study showed that the TA-HQ/*p*-PDA system has an unclear glass transition on DMA, an ultralow CTE (3.2 ppm K^−1^), and a very high tensile modulus (8.9 GPa)] [11], which are very similar to those of the PMDA/AB-HQ system. Similarly, there was almost no difference in these properties between A-type and B-type PEsIs when including the small substituents (–CH_3_ and –OCH_3_ groups), as shown in Figure 8. On the other hand, the introduction of the phenyl-containing HQDB unit to the tetracarboxylic dianhydride structure caused a seemingly significant decrease in *T*_g_ (250 °C) that probably corresponds to an intensified sub-glass transition [16,34]. This is based on the disturbed intermolecular dipole-dipole interaction between the imide C=O groups [13,35,36] caused by the presence of the bulky phenyl side groups. Thus, the introduction of the phenyl-containing HQDB unit to the tetracarboxylic dianhydride structure involved a higher risk of property deterioration than when it was introduced to the diamine structure.

On the hand, the water absorption behavior clearly depends on the position of the HQDB unit (whether A-type or B-type); non-substituted PMDA/AB-HQ (**#1**) is superior to the corresponding TA-HQ/*p*-PDA system in terms of suppressing the *W*_A_ (0.86% for the former and 1.6% for the latter [11]). A similar effect on the *W*_A_ was also observed in a comparison of the methyl-substituted counterparts.

#### 3.5.3. Correlation of *W*_A_ and Imide Group Content

Figure 9 shows the impact of imide group content (*C*_i_) on the *W*_A_ for the PEsIs obtained using a series of the ester-linked diamines together with other conventional PEsIs and commercially available PI films (KAPTON-H [26] and UPILEX-S [37]). The results indicate that the use of ester-linked diamines, developed in this study, are very effective in reducing the *W*_A_, as suggested from the plots (●) that were located in the lower left region in Figure 9. A rough correlation, where the *W*_A_ decreases with decreasing *C*_i_, is observed in this figure, which suggests that the highly polarized imide groups as the chemical factor participate strongly in the water absorption phenomenon of these PEsI films. This also means that the ester C=O groups do not significantly contribute to enhancing the *W*_A_ as the imide groups can be reduced by incorporating the ester-linked aromatic units to the main chains (by increasing the ester group content). However, no matter how thoroughly the *C*_i_ was reduced, the *W*_A_ value is unlikely to become null as the *W*_A_–*C*_i_ curve seems to asymptotically approach a low but non-zero *W*_A_ value (~ 0.1–0.2%) as *C*_i_ is decreased. This prediction of the existence of a lower limit of *W*_A_ in the present PEsI systems does not conflict with the fact that an aromatic polyester (polyarylate) system without imide groups has a low but non-zero *W*_A_ value (0.26% [38]), which approximates the above-mentioned lower limit of *W*_A_. Figure 9 also shows that some of the plots are significantly scattered. This suggests that the *W*_A_ values of the PEsIs are also affected somewhat by a physical factor such as crystallinity, as discussed in our previous study [14], corresponding to a reasonable assumption that water molecules do not diffuse and penetrate easily into the crystal phases. The prominent effect of crystallinity on suppressing water uptake is also observed in poly(ethylene terephthalate) systems [39]. 

#### 3.5.4. Correlation of CHE and *W*_A_

Figure 10 depicts the influence of *W*_A_ on CHE for the PEsIs obtained using the ester-linked diamines, together with other conventional PEsIs and commercially available PI films [26,37]. The present PEsI systems had very low CHE values, as suggested from the plots (●, ○) that were located in the lower left region in this figure. The rough correlation, observed between CHE and *W*_A_, suggests that the water absorption behavior is the primary factor governing CHE. In addition, CHE can also be influenced by the degree of in-plane orientation as the secondary factor, as discussed previously [14].

#### 3.5.5. Correlation of Tensile Modulus and CTE

A relatively good correlation was observed between the tensile modulus and CTE for the PEsIs derived from the ester-linked diamines, as shown in Figure 11. This is reasonable because tensile modulus and CTE are both dominated by the same factors, i.e., the degree of in-plane orientation and chain rigidity. If there were a poor correlation between them, it is likely that the CTE values were measured under inappropriate conditions; the TMA curves often undergo significant deformation when the specimens include residual strain and/or adsorbed water, whereby inaccurate CTE data are obtained [40]. 

In some cases, FPCs are used in a folded state in narrow spaces of electronic devices. The dielectric films, used here, make a low CTE and a lower modulus desired simultaneously. The latter is required to suppress a spring-back force of the folded FPCs. However, an enhancement of the modulus is, in principle, inevitable with a decrease in CTE, as shown in Figure 11. Copolymerization using 4,4′-ODA seems to be somewhat effective in reducing the modulus, while maintaining a low CTE property, as suggested from the plots (○) in this figure. Our attempts at overcoming the trade-off between low modulus and low CTE properties will be reported elsewhere [41].

### 3.6. Flame Retardancy

The results of the UL-94V test are summarized in Table 8. We have reported previously that the introduction of thermally less stable substituents, such as methyl groups into ester-linked tetracarboxylic dianhydrides results in a significant deterioration in flame retardancy [14]. From this point of view, the substituent-free AB-HQ-based PEsIs are expected to have good flame retardancy. Unfortunately, the AB-HQ-based PEsI copolymers (**#4, #5**) showed poor flame retardancy; all five specimens burned up to the clamped top edge on the UL-94V test. On the other hand, unexpectedly, the methoxy-substituted PMDA/AB-MeOHQ system (**#9**), as well as conventional wholly aromatic PI films [26,37], passed the V-0 standard, which represents the highest flame retardancy. These results are probably attributable to the oxygen blocking effect originating from the formation of char layers [42,43]. In contrast, the use of TA-HQ (**#11**) instead of PMDA caused a significant deterioration in flame retardancy, thus suggesting that the increase in the ester groups is disadvantageous for obtaining excellent flame retardancy. On the other hand, the combination of s-BPDA and AB-MeOHQ (**#10**) was close to passing the V-0 standard as three specimens (of five) survived without burning up to the clamped top edge. It is highly possible that this PEsI attained the V-0 standard either by a slight increase in specimen thickness or copolymerization using a miniscule amount of a phosphorus-containing monomer [14,44]. The copolymer using 4,4′-ODA (**#12**) also passed the V-0 standard, despite the presence of thermally less stable methoxy groups in the structure. On the other hand, against our prediction, the AB-14DHN-based copolymer (**#20**), including a condensed benzene side group, showed poor flame retardancy. In contrast, the AB-DP44BP-based copolymer (**#27**) with thermally stable phenyl substituents passed the V-0 standard, with the shortest burning time among the tested samples.

The fact that the s-BPDA/AB-MeOHQ;4,4′-ODA copolymer with the thermally less stable methoxy substituent demonstrated excellent flame retardancy and attracted our interests. This mechanism is under investigation.

### 3.7. Dielectric Properties

The suppression of the transmission loss of electronic signals in circuit boards has been a crucial subject in recent high-speed mobile communication systems, operating at high frequencies. The transmission loss (*α*_tl_) is expressed as,
*α*_tl_ = *α*_d_ + *α*_c_(3)
where *α*_d_ and *α*_c_ denote the dielectric loss and conductor loss, respectively. The former is represented by Equation (4),
*α*_d_ = 27.3 (*f*/*c*) *ε*_r_^0.5^ tan *δ*(4)
where *f* is the operating frequency (in Hz) and *c* is the speed of light (in m/s). This relationship also suggests that a direct strategy for suppressing the *α*_d_ is to reduce the *ε*_r_ and tan *δ* of the dielectric substrates, particularly the latter [4]. The PEsI film (**#10**) showed a moderate *ε*_r_ (3.18) and a low tan *δ* (3.14 × 10^−3^) at 10 GHz in 50% RH and 23 °C, values that approximate those of a commercially available liquid-crystalline polyester film (e.g., *ε*_r_ = 3.3 and tan *δ* = 2 × 10^−3^ at 28 GHz and 25 °C for a 50 μm-thick Vecstar^TM^ film, Kraray Co. [45]). Thus, the PEsI system (**#10**) is also suitable for suppressing dielectric loss.

### 3.8. Performance Balance of the PEsIs for Use in High-performance FPCs

The performance balance of the PEsIs was visualized using a spider chart, which can be regarded as having a good balance when spread evenly. Each target property was ranked into five groups on the basis of the criteria for use in high-performance FPCs (Table 9), as established in our previous paper [14]; for example, when a material has a very low CTE (CTE < 10 ppm K^−1^), the low CTE property of the material was assigned as “rank-5”. This assignment is based on a typical example of low-CTE PI systems (e.g., CTE = 5–15 ppm K^−1^ for s-BPDA/*p*-PDA [24,25]). On the other hand, high-CTE materials (CTE > 70 ppm K^−1^), as generally observed in common flexible PI systems, were assigned as “rank-1”. The intermediate ranks (2–4) were established by dividing equally the CTE difference between the upper (70 ppm K^−1^) and lower limits (10 ppm K^−1^), as listed in Table 9.

Figure 12 indicates the performance balance for the PEsI (**#10**) as an example, together with the appearance of the film. A relatively well-spread spider chart was obtained for this system, although there is room for further improvement in toughness and *ε*_r_. The results suggest that this system is a promising candidate as the dielectric substrate material for use in the next generation of high-performance FPCs.

## 4. Conclusions

A series of ester-linked diamines, with different lengths and substituents, was synthesized in this study. AB-HQ without substituents was poorly soluble in NMP at room temperature. PAA polymerization was carried out by adding tetracarboxylic dianhydride solid into a hot diamine solution to ensure the initial solution homogeneity. This procedure enabled smooth progress of the polymerization, although it was accompanied by a significant decrease in the η_inh_ value, particularly when using hydrolytically less stable PMDA. On the other hand, the polyaddition of AB-HQ and s-BPDA in a similar manner led to a homogeneous solution of PAA with a sufficiently high η_inh_ value, probably owing to the higher hydrolytic stability of s-BPDA. The modification of the s-BPDA/AB-HQ system by copolymerization using 4,4′-ODA (40 mol%) significantly improved the film toughness (*ε*_b_^max^ = 47%), while maintaining a low CTE value (11.7 ppm K^−1^), a relatively low *W*_A_ (0.64%), a very low CHE (4.5 ppm/RH%), and a relatively high *T*_d_^5^ (524 °C in N_2_). Unfortunately, this copolymer failed to achieve the highest flame retardancy (UL-94, V-0 standard).

On the other hand, the incorporation of various substituents (–CH_3_, –OCH_3_, and phenyl groups) to AB-HQ was very effective in improving diamine solubility, which enabled the application of the simple polymerization process without the initial heating of the diamine solutions and easily led to PAAs of sufficiently high molecular weights. The effects of substituent bulkiness on the target properties were also discussed. The introduction of bulkier phenyl substituent tended to increase the CTE of the PEsI films, in contrast to that of the small substituents (–CH_3_, –OCH_3_). 

We also used AB-44BP and AB-MHQHB, consisting of longitudinally further extended structures, with expectations for further improvement in *W*_A_ and CHE, while maintaining low CTE property. However, this approach was unsuccessful; these diamines were poorly soluble in NMP at room temperature, so that the polyaddition was carried out while heating the diamine solutions, and resulted in low molecular weights of the PAAs. Consequently, the CTE values of the resultant PEsI films were not as low as we expected initially.

The correlations between some important parameters (*W*_A_ vs. *C*_i_ and CHE vs. *W*_A_) were also investigated. From these correlations, the dominant factors of *W*_A_ and CHE were discussed in this study.

Among the PEsIs derived from the AB-HQ analogs, including various substituents, the methoxy-substituted s-BPDA/AB-MeOHQ system exhibited well-balanced properties having very high *T*_g_, low CTE, low CHE, low *W*_A_ values, and sufficient ductility. Although, the 26 μm-thick film narrowly missed certification of the V-0 standard in a UL-94V test. The s-BPDA/AB-MeOHQ film also displayed a moderate *ε*_r_ (3.18) and a low tan *δ* (3.14 × 10^−3^) at 10 GHz under 50% RH and 23 °C. Thus, this PEsI system is a promising candidate as a group of novel dielectric substrate materials for use in the next generation of high-performance FPCs operating at higher frequencies (≥10 GHz).

## Figures and Tables

**Figure 1 polymers-12-00859-f001:**
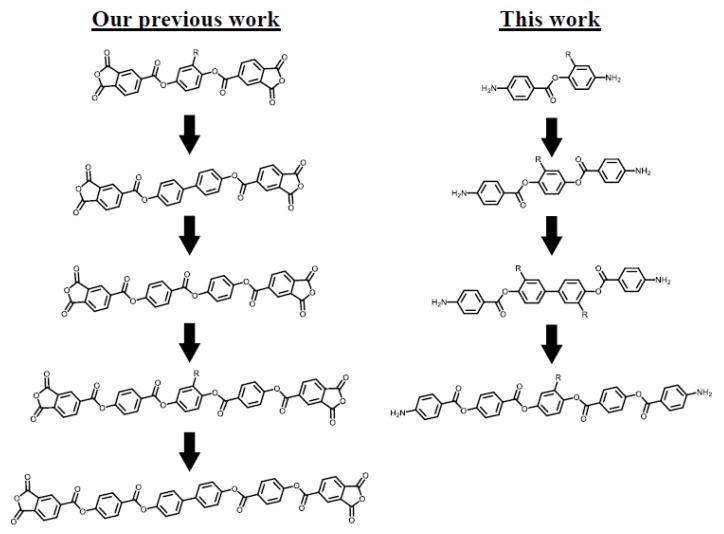
Our strategies for obtaining the next-generation of high-temperature dielectric substrate materials in FPCs by using ester-linked monomers with longitudinally extended structures.

**Figure 2 polymers-12-00859-f002:**
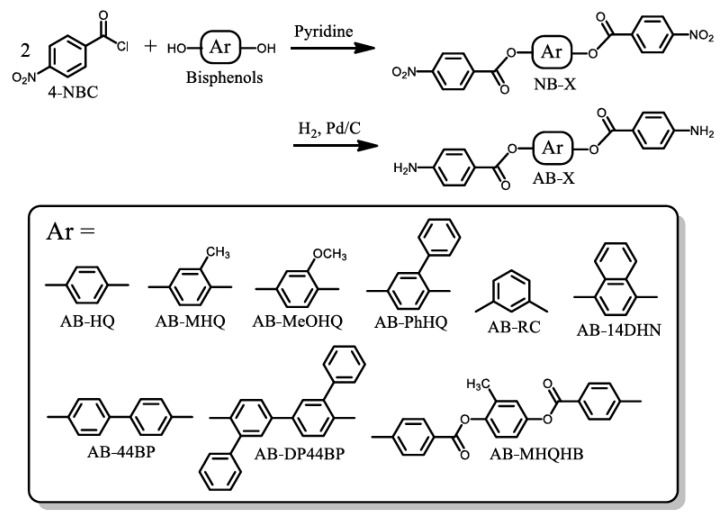
Reaction scheme for the synthesis of ester-linked diamines with different substituents.

**Figure 3 polymers-12-00859-f003:**
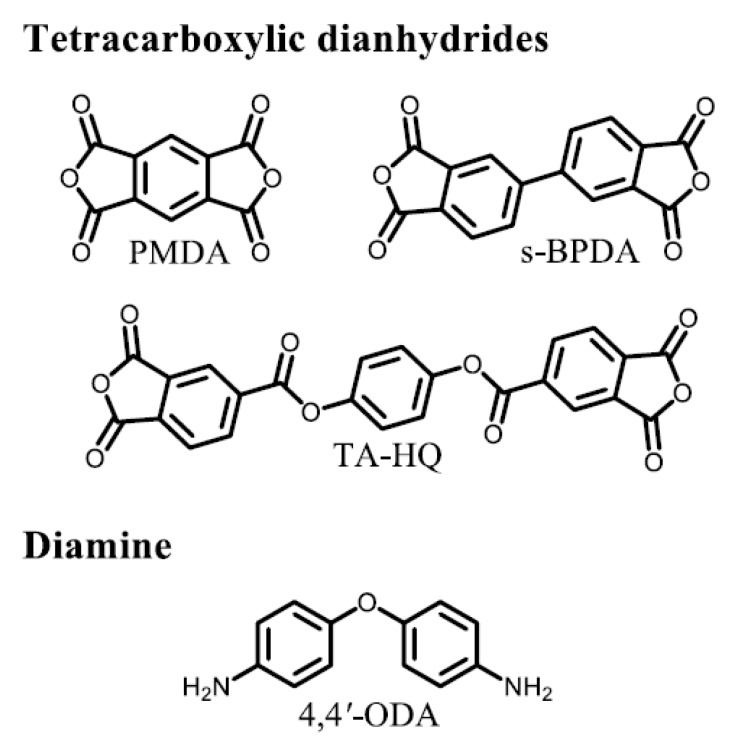
Structures and abbreviations of the common monomers used in this study.

**Figure 4 polymers-12-00859-f004:**
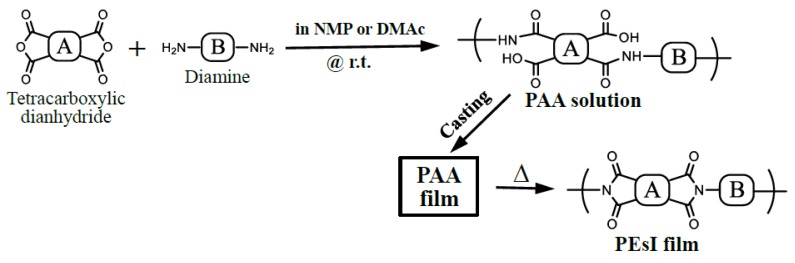
Schemes of polyaddition, solution casting of PAAs, and thermal imidization for PEsI film preparation.

**Figure 5 polymers-12-00859-f005:**
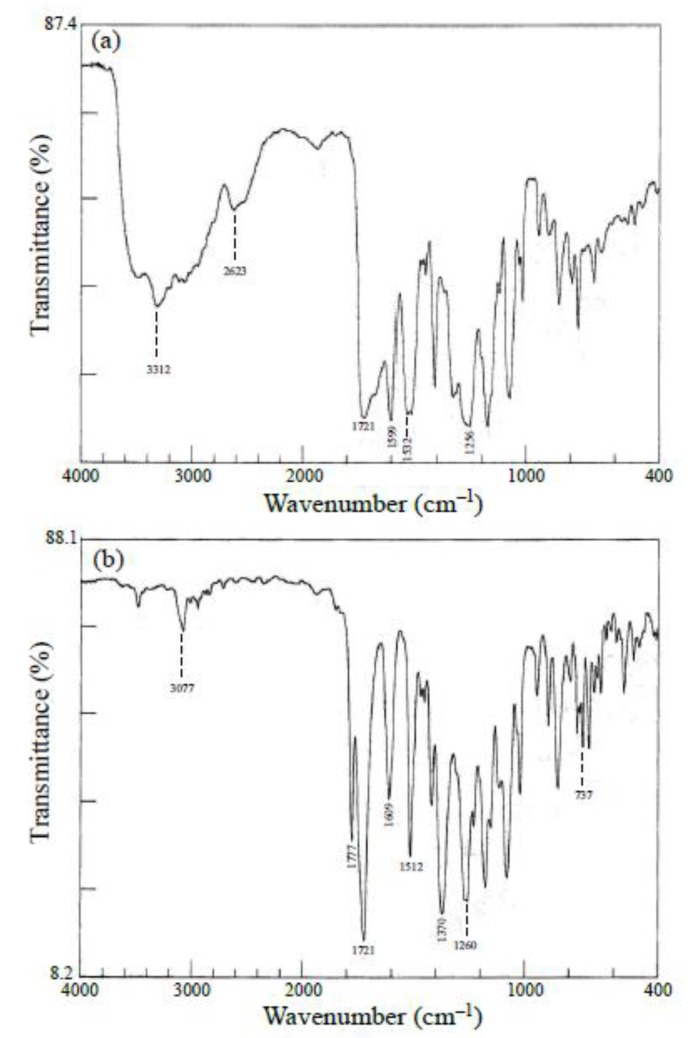
FT-IR spectra of thin films for the s-BPDA/AB-MeOHQ system: (**a**) PAA and (**b**) PEsI.

**Figure 6 polymers-12-00859-f006:**
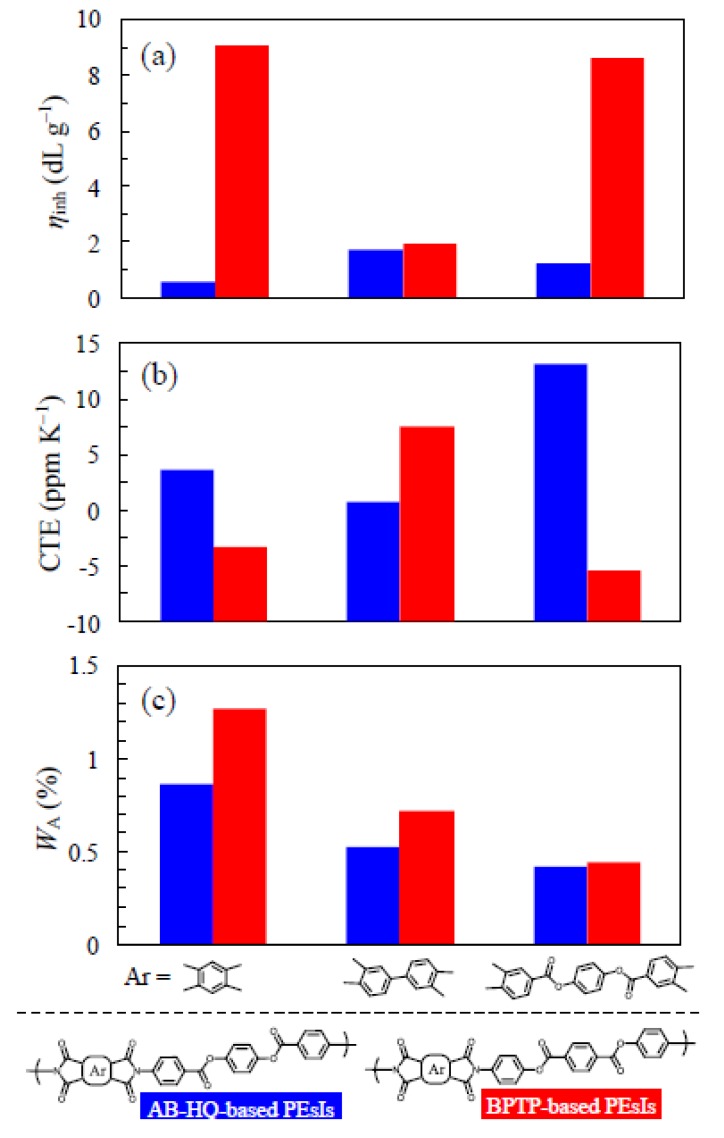
Comparisons of the properties for the AB-HQ-based (left/blue bar) and BPTP-based systems (right/red bar): (**a**) *η*_inh_ of PAAs, (**b**) CTE, and (**c**) water uptake of the PEsI films.

**Figure 7 polymers-12-00859-f007:**
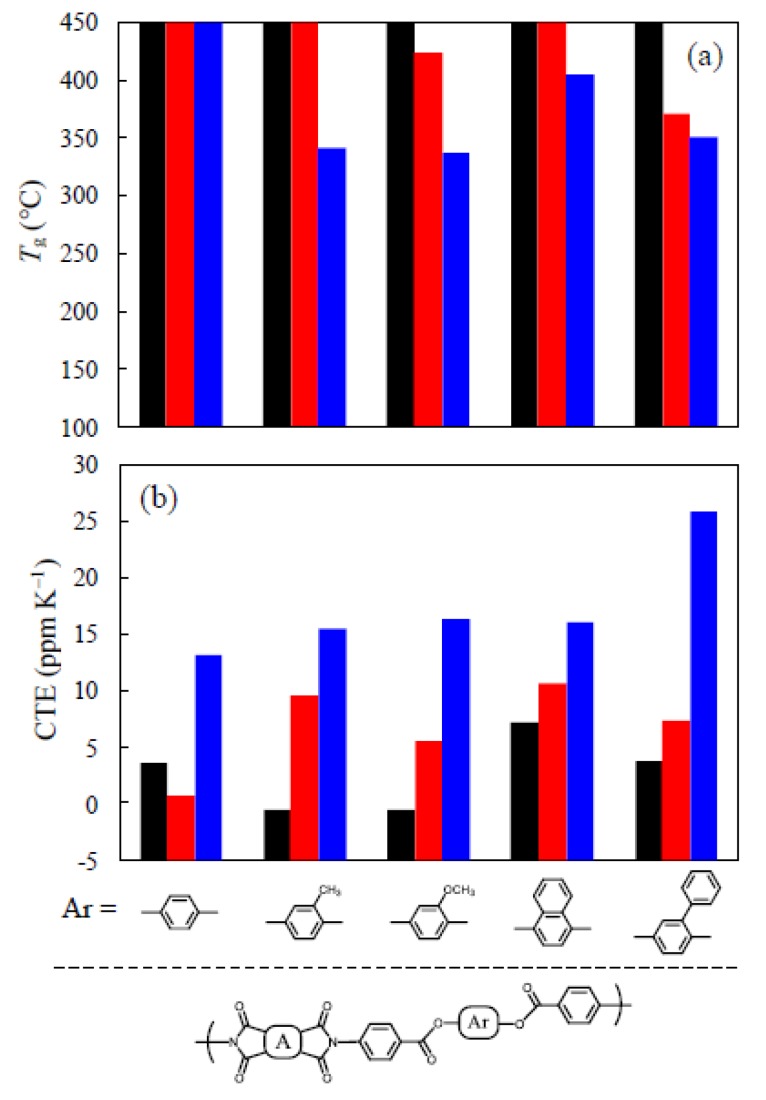
Influence of side group bulkiness on *T*_g_ (**a**) and CTE (**b**) for PEsIs derived from AB-HQ analogs and different tetracarboxylic dianhydrides: PMDA [left (black) bars], s-BPDA [center (red) bars)], and TA-HQ [right (blue) bars]. When no distinct *T*_g_ was observed up to 450 °C on DMA, the *T*_g_ was represented as being higher than 450 °C in this graph.

**Figure 8 polymers-12-00859-f008:**
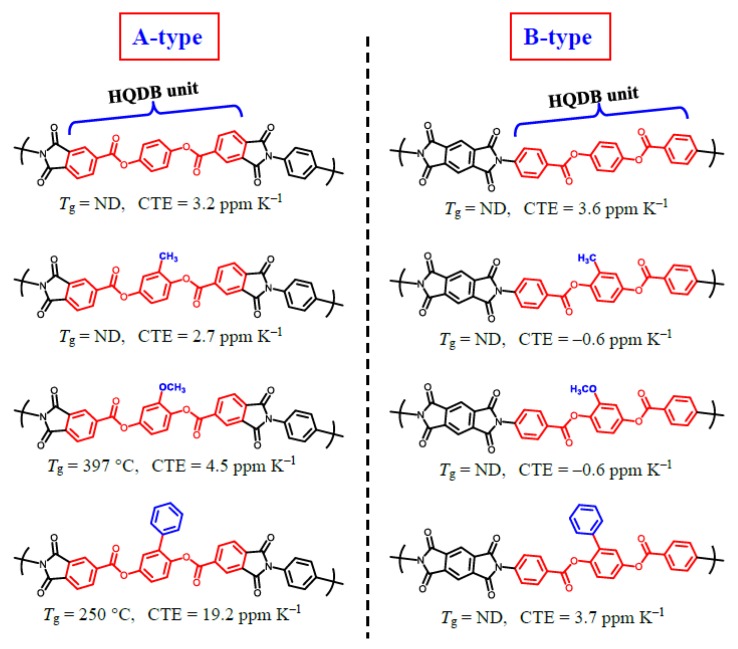
Comparisons of *T*_g_ and CTE between the PEsIs (A-type) obtained from tetracarboxylic dianhydrides including hydroquinone dibenzoate (HQDB) unit (red-marked) with different substituents (blue-marked) and the PEsIs (B-type) obtained from HQDB-containing diamines.

**Figure 9 polymers-12-00859-f009:**
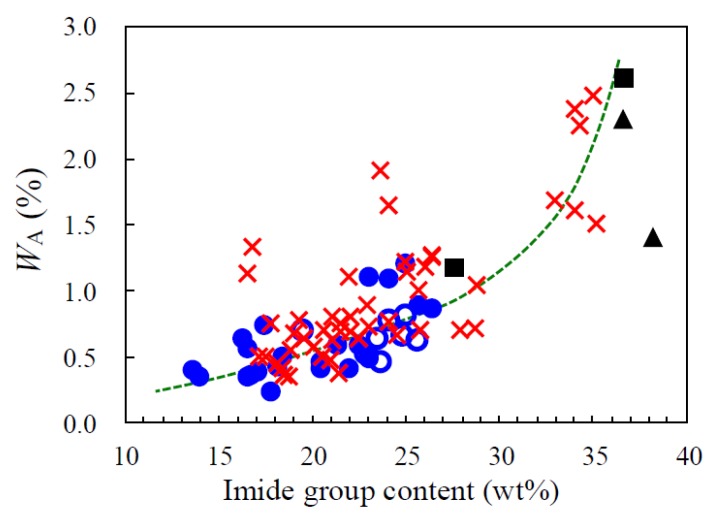
Relationship between the imide content and the extent of water absorption for PEsI or PI films: (●) homo PEsIs developed in this work, (○) the related PEsI copolymers, (**×**) conventional PEsIs that we have examined previously, (■) PMDA-based conventional PIs, and (▲) commercially available PI films. The data for fluorine- and amide-containing PEsIs are not included in this figure.

**Figure 10 polymers-12-00859-f010:**
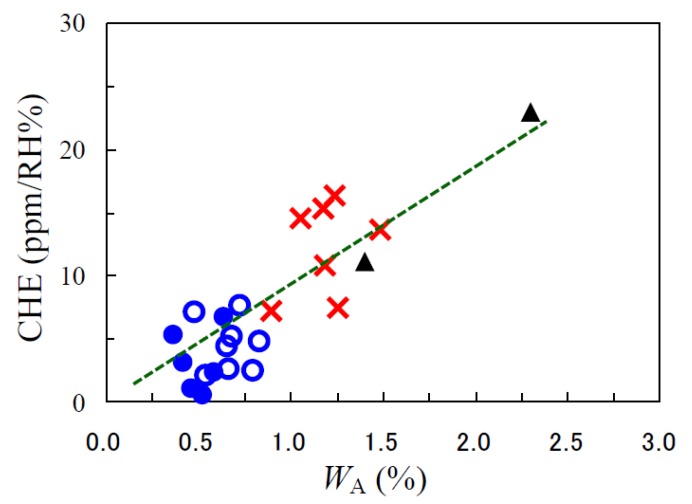
Correlation of the CHE and *W*_A_ for PEsI or PI films: (●) homo PEsIs developed in this work, (○) the related PEsI copolymers, (**×**) conventional PEsIs that we have examined previously, and (▲) commercially available PI films. The data for fluorine- and amide-containing PEsIs are not included in this figure.

**Figure 11 polymers-12-00859-f011:**
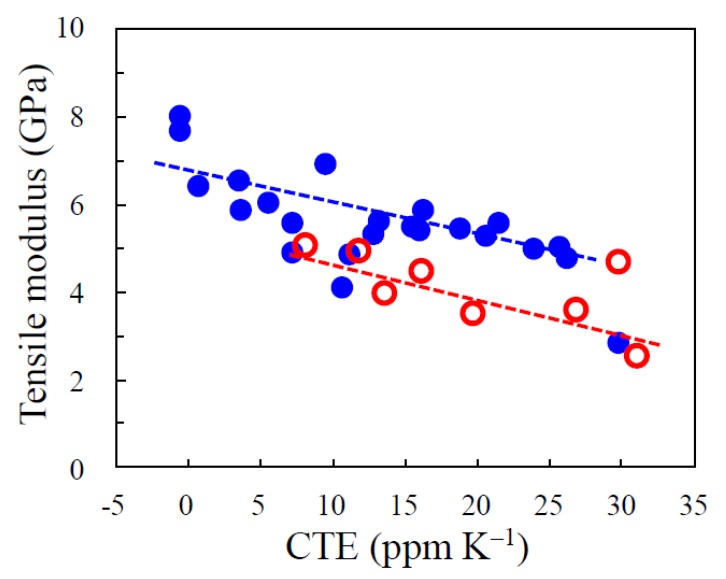
Correlation of the tensile modulus and CTE for the PEsIs developed in this work: (●) homo PEsIs and (○) the related copolymers.

**Figure 12 polymers-12-00859-f012:**
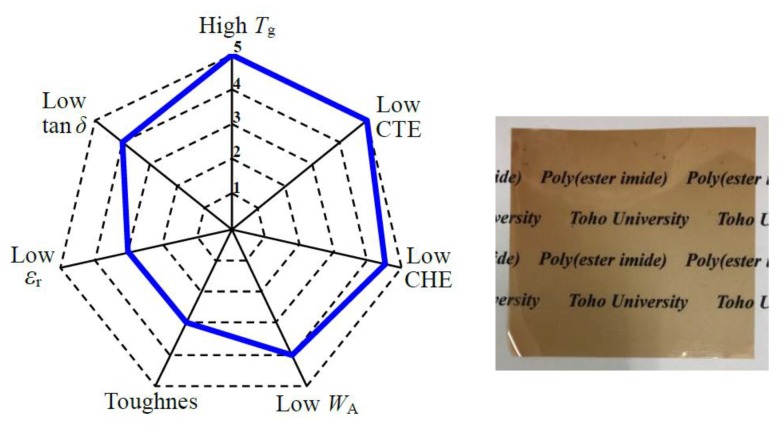
Performance balance for the s-BPDA/AB-MeOHQ system (**#10**) on the basis of the five-rank criteria (Table 9) and the appearance of the film.

**Table 1 polymers-12-00859-t001:** Film properties of the PEsIs derived from AB-HQ and common tetracarboxylic dianhydrides (TCDA) and related copolymer.

No.	Diamine (mol %)	TCDA	*η*_inh_ PAA (dL g^−1^)	*T*_g_ (°C)	CTE (ppm K^−1^)	*E* (GPa)	*ε*_b_ Ave/Max (%)	*σ*_b_ (GPa)	*T*_d_^5^ in N_2_ (°C)	*T*_d_^5^ in air (°C)	*W*_A_ (%)	CHE (ppm/RH%)
1	AB-HQ	PMDA	0.57	ND ^a^	3.6	6.52	7.7/9.8	0.227	526	483	0.86	----
2	ibid	s-BPDA	1.72	ND	0.7	6.41	5.8/7.4	0.211	530	489	0.52	0.5
3	ibid	TA-HQ	1.21	ND	13.2	5.61	15.3/20.1	0.207	498	478	0.42	----
4	AB-HQ (70) 4,4′-ODA (30)	s-BPDA	1.85	ND	8.0	5.11	23.1/38.4	0.258	528	507	0.82	4.9
5	AB-HQ (60) 4,4′-ODA (40)	s-BPDA	1.16	ND	11.7	4.99	35.8/47.1	0.296	524	503	0.64	4.5

^a^ ND: Not detected by DMA.

**Table 2 polymers-12-00859-t002:** Film properties of the PEsIs derived from AB-MHQ (R = –CH_3_) with tetracarboxylic dianhydrides (TCDA).

No.	Diamine	TCDA	*η*_inh_ PAA (dL g^−1^)	*T*_g_ (°C)	CTE (ppm K^−1^)	*E* (GPa)	*ε*_b_ Ave/Max (%)	*σ*_b_ (GPa)	*T*_d_^5^ in N_2_ (°C)	*T*_d_^5^ in air (°C)	*W*_A_ (%)	CHE (ppm/RH%)
6	AB-MHQ	PMDA	1.21	ND ^a^	–0.6	8.00	9.2/12.8	0.327	492	464	0.88	----
7	ibid	s-BPDA	1.16	ND	9.5	6.91	3.4/4.6	0.181	498	473	0.58	2.3
8	ibid	TA-HQ	1.53	342	15.5	5.46	26.4/34.4	0.242	476	442	0.23	----

^a^ ND: Not detected by DMA.

**Table 3 polymers-12-00859-t003:** Film properties of the PEsIs derived from AB-MeOHQ (R = –OCH_3_) with tetracarboxylic dianhydrides (TCDA) and related copolymer.

No.	Diamine (mol %)	TCDA	*η*_inh_ PAA (dL g^−1^)	*T*_g_ (°C)	CTE (ppm K^−1^)	*E* (GPa)	*ε*_b_ Ave/Max (%)	*σ*_b_ (GPa)	*T*_d_^5^ in N_2_ (°C)	*T*_d_^5^ in air (°C)	*W*_A_ (%)	CHE (ppm/RH%)
9	AB-MeOHQ	PMDA	1.57	ND ^a^	–0.6	7.68	4.7/6.2	0.213	467	451	1.20	----
10	ibid	s-BPDA	1.36	424	5.6	6.01	15.1/23.0	0.236	473	458	0.41	3.1
11	ibid	TA-HQ	2.05	338	16.3	5.85	2.5/3.1	0.129	444	427	0.73	----
12	AB-MeOHQ (70)4,4′-ODA (30)	s-BPDA	1.58	430	13.5	4.01	16.5/27.7	0.197	470	461	0.78	2.5

^a^ ND: Not detected by DMA.

**Table 4 polymers-12-00859-t004:** Film properties of the PEsIs derived from AB-PhHQ (R = phenyl group) with tetracarboxylic dianhydrides (TCDA) and related copolymer.

No.	Diamine (mol %)	TCDA	*η*_inh_ PAA (dL g^−1^)	*T*_g_ (°C)	CTE (ppm K^−1^)	*E* (GPa)	*ε*_b_ Ave/Max (%)	*σ*_b_ (GPa)	*T*_d_^5^ in N_2_ (°C)	*T*_d_^5^ in air (°C)	*W*_A_ (%)	CHE (ppm/RH%)
13	AB-PhHQ	PMDA	1.12	ND ^a^	3.7	5.87	6.9/9.3	0.239	491	446	1.10	----
14	ibid	s-BPDA	0.92	371	7.3	4.89	2.0/2.5	0.091	488	456	0.46	1.0
15	ibid	TA-HQ	1.25	352	25.8	5.01	13.5/22.2	0.215	478	445	0.34	----
16	AB-PhHQ (70) 4,4′-ODA (30)	s-BPDA	1.15	326	29.7	4.71	7.7/10.3	0.206	484	463	0.53	2.2

^a^ ND: Not detected by DMA.

**Table 5 polymers-12-00859-t005:** Film properties of the PEsIs derived from AB-14DHN with tetracarboxylic dianhydrides (TCDA) and related copolymer.

No.	Diamine (mol %)	TCDA	*η*_inh_ PAA (dL g^−1^)	*T*_g_ (°C)	CTE (ppm K^−1^)	*E* (GPa)	*ε*_b_ Ave/Max (%)	*σ*_b_ (GPa)	*T*_d_^5^ in N_2_ (°C)	*T*_d_^5^ in air (°C)	*W*_A_ (%)	CHE (ppm/RH%)
17	AB-14DHN	PMDA	1.51	ND ^a^	7.2	5.58	2.4/2.7	0.154	492		1.09	----
18	ibid	s-BPDA	1.24	ND	10.7	4.09	1.8/2.2	0.103	488	422	0.58	----
19	ibid	TA-HQ	1.12	>405	16.1	5.41	5.5/8.6	0.194	469	451	0.38	----
20	AB-14DHN (70) 4,4′-ODA (30)	s-BPDA	1.28	ND	16.0	4.51	6.9/10.9	0.199	471	456	0.65	2.7

^a^ ND: Not detected by DMA.

**Table 6 polymers-12-00859-t006:** Film properties of the PEsIs derived from AB-44BP with tetracarboxylic dianhydrides (TCDA).

No.	TCDA	*η*_inh_ PAA (dL g^−1^)	*T*_g_ (°C)	CTE (ppm K^−1^)	*E* (GPa)	*ε*_b_ Ave/Max (%)	*σ*_b_ (GPa)	*T*_d_^5^ in N_2_ (°C)	*T*_d_^5^ in air (°C)	*W*_A_ (%)
21	PMDA	0.51	ND ^a^	26.3	4.75	3.5/4.4	0.121	518	477	0.48
22	s-BPDA	0.89	ND	18.8	5.44	4.4/6.3	0.151	512	491	0.41
23	TA-HQ	0.59	ND	20.6	5.28	5.5/7.3	0.153	483	464	0.56

^a^ ND: Not detected by DMA.

**Table 7 polymers-12-00859-t007:** Film properties of the PEsIs derived from AB-DP44BP (R = phenyl group) with tetracarboxylic dianhydrides (TCDA).

No.	Diamine (mol %)	TCDA	*η*_inh_ PAA (dL g^−1^)	*T*_g_ (°C)	CTE (ppm K^−1^)	*E* (GPa)	*ε*_b_ Ave/Max (%)	*σ*_b_ (GPa)	*T*_d_^5^ in N_2_ (°C)	*T*_d_^5^ in air (°C)	*W*_A_ (%)	CHE (ppm/RH%)
24	AB-DP44BP	PMDA	2.05	ND ^a^	11.2	4.83	6.6/8.7	0.187	499	415	0.49	----
25	ibid	s-BPDA	1.07	433	12.8	5.29	3.7/4.4	0.169	501	461	0.36	5.2
26	ibid	TA-HQ	1.28	322	24.0	4.99	9.5/16.6	0.204	479	433	0.35	----
27	AB-DP44BP (80) 4,4′-ODA (20)	s-BPDA	1.02	372	31.0	2.58	6.9/8.5	0.116	508	417	0.71	7.7

^a^ ND: Not detected by DMA.

**Table 8 polymers-12-00859-t008:** Flame retardancy of the PEsIs derived from AB-Xs and rigid tetracarboxylic dianhydrides and related copolymer.

No.	Diamines (mol %)	Tetracarboxylic Dianhydrides	UL-94, V-0 (*d*)
4	AB-HQ (70) 4,4′-ODA (30)	s-BPDA	Failed ^a^ (20 μm)
5	AB-HQ (60) 4,4′-ODA (40)	s-BPDA	Failed ^a^ (24 μm)
9	AB-MeOHQ	PMDA	Passed (19 μm)
10	AB-MeOHQ	s-BPDA	Failed ^b^ (26 μm)
11	AB-MeOHQ	TA-HQ	Failed ^a^ (31 μm)
12	AB-MeOHQ (70) 4,4′-ODA (30)	s-BPDA	Passed (24 μm)
20	AB-14DHN (70) 4,4′-ODA (30)	s-BPDA	Failed ^a^ (25 μm)
27	AB-DP44BP (80) 4,4′-ODA (20)	s-BPDA	Passed (29 μm)

^a^ All five specimens (out of five) burned up to the clamped top edge. ^b^ Two specimens (out of five) burned up to the clamped top edge.

**Table 9 polymers-12-00859-t009:** Criteria for estimating the performance balance of the PEsIs for use in the dielectric substrates of high-performance FPCs.

Properties	Parameters	Relative Rank				
		1	2	3	4	5
Heat resistance	*T*_g_ (°C)	<210	220–250	260–290	300–330	>350 or ND ^a^
Low thermal expansion property	CTE (ppm K^−1^)	>70	60–50	45–35	30–20	<10
Low hygroscopic expansion property	CHE (ppm/RH%)	>30	25–20	15–10	8–4	<2
Low water absorption	*W*_A_ (%)	>3.0	2.5–2.0	1.5–1.0	0.8–0.4	<0.2
Toughness	*ε*_b_^max^ (%)	No film-forming ability or <2	5–10	20–30	40–60	>80
Low dielectric constant ^b^ in GHz range	*ε* _r_	>3.7	3.6–3.4	3.3–3.1	3.0–2.8	<2.7
Low dissipation factor in GHz range ^b^	tan *δ*	>0.02	0.01–0.008	0.007–0.005	0.004–0.002	<0.001

^a^ Non-detected *T*_g_ up to 450 °C on DMA. ^b^ Data measured under ambient conditions (40–60 RH % and 20–30 °C).

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
