# Peer review of "Poly(ester imide)s Possessing Low Coefficients of Thermal Expansion and Low Water Absorption (V). Effects of Ester-linked Diamines with Different Lengths and Substituents"

_polymers, 2020, doi:10.3390/polym12040859_

Round 1

Reviewer 1 Report

It is an excellent paper about poly(ester imide)s having low CTE and low water absorption synthesized from ester-linked diamines. Many poly(ester imide)s were systematically investigated. I feel that the scientific value of this paper is sufficient to justify immediate publication without change.

Author Response

Thank you for your good evaluations for our manuscript.

Reviewer 2 Report

Polymers

Manuscript Number:  polymers-767519

Title: "Poly(ester imide)s Possessing Low Coefficients of Thermal Expansionand Low Water Absorption (V). Effects of Ester-Linked Diamines with DifferentLengths and Substituents"

Author(s): Masatoshi Hasegawa and Tomoaki Hishiki 

The current paper describes the targeted preparation of novel ester-linked diamines with different lengths and substituents, then, based on these diamines the authors describe the synthesis of a series of new poly(ester imide)s having improved high performance properties. The introduction of various substituents into the structure of the diamines resulted in improved solubility in the final products. Also, increased coefficients of thermal expansion, and desired water uptake and coefficients of hygroscopic expansion were achieved by structural design. 

The practical application potential providing a feasible way for the development of novel poly(ester imide)s with high performance properties, such as, thermal stability, fire resistance, dielectric constant, is discussed. 

The chemical structure of prepared compounds is rigorously confirmed by specific methodologies (FTIR, RMN). Thermal stability, mechanical properties and flame retardancy of epoxies were evaluated by TGA, DMA, tensile testing, and L-94 test.  

The paper is well written and the characterization techniques are logically chosen, the scientific subject is of actuality and it is related to the industry, from practical point of view. The current work is qualified for publishing in Polymers.  

Author Response

(The authors gave the same response as above.)

Reviewer 3 Report

This manuscript introduces  a good contribution in this scientific field.

Before accepting, it should be explain in detail:

1) Why did the authors not use e.g. lithium chloride to suppress a polyelectrolyte effect in viscometric measurements?

2) Why was the glass transition temperature  not detected for some materials by a DMA (it would be appropriate to show at least on such record)?

3) Is it a specific scientific issue that a co-author of almost a third of literary references must be dr. Hasegawa (and in addition to that: 31. Hasegawa M.,....32. M. Hasegawa...)?  

Author Response

Reviewer-3

This manuscript introduces  a good contribution in this scientific field.

Before accepting, it should be explain in detail:

1) Why did the authors not use e.g. lithium chloride to suppress a polyelectrolyte effect in viscometric measurements?

The use of LiCl may suppress the polyelectrolyte effect for poly(amic acid)s (PAAs) and allow the determination of inherent viscosities. However, we did not dare to use LiCl in our viscosity measurements because this effect for PAAs is generally negligible at 0.5 wt% at 30 degree C even in the absence of LiCl, although it becomes prominent at much lower concentrations (< 0.1 wt%). Therefore, as well as most of previously reported papers dealing with PAAs, we used the reduced viscosity data measured at 0.5 wt% at 30 degree C in this paper rather than the inherent viscosities determined from extrapolation to zero concentration.

2) Why was the glass transition temperature not detected for some materials by a DMA (it would be appropriate to show at least on such record)?

Indeed, some polyimide systems particularly having rod-like main chain structures show no-Tg behavior on DMA up to 450 degree C, despite the fact that DMA is the most sensitive method for detecting Tg. This may be attributed to the fact that the Tgs exist above the upper limit temperature of the DMA (450 degree C). Another possibility is that appreciable softening behavior does not occur practically even above the Tg in such rod-like systems because of expected higher crystallinity and local ordering, and poor molecular motions.

3) Is it a specific scientific issue that a co-author of almost a third of literary references must be dr. Hasegawa (and in addition to that: 31. Hasegawa M.,....32. M. Hasegawa...)?  

I understand that your point is that too many references from our research group are cited in this manuscript. However, actually, we have no intention of increasing the citations of our previous papers on purpose. To cite many of our previous papers was inevitable to discuss the present main subject, i.e., the development of low-CTE and low-CHE poly(ester imide)s for use in the next generation of FPC from the academic points of view, which have not been extensively discussed in other groups.